# Visual Prognostic Factors in Eyes with Subretinal Fluid Associated with Branch Retinal Vein Occlusion

**DOI:** 10.3390/jcm12082909

**Published:** 2023-04-17

**Authors:** Hirofumi Sasajima, Masahiro Zako, Kenta Murotani, Hidetoshi Ishida, Yoshiki Ueta, Naoko Tachi, Takafumi Suzuki, Yuji Watanabe, Yoshihiro Hashimoto

**Affiliations:** 1Department of Ophthalmology, Shinseikai Toyama Hospital, Imizu 939-0243, Japan; 2Yamada Eye Clinic, Nagano 380-0813, Japan; 3Department of Ophthalmology, Asai Hospital, Seto 489-0866, Japan; 4Biostatistics Center, Kurume University, Kurume 830-0011, Japan; 5Department of Ophthalmology, Kanazawa Medical University, Kahoku 920-0293, Japan; 6Tachi Eye Clinic, Toyama 930-0002, Japan; 7Department of Ophthalmology, University of Tokyo Hospital, Tokyo 113-8655, Japan

**Keywords:** branch retinal vein occlusion, ellipsoid zone, macular edema, optical coherence tomography, subretinal fluid, visual prognostic factor

## Abstract

We investigated whether foveal ellipsoid zone (EZ) status affects visual prognosis in eyes with subretinal fluid (SRF) associated with branch retinal vein occlusion (BRVO). We included 38 eyes retrospectively and classified those with or without a continuous EZ on the SRF of the central foveola on the vertical optical coherence tomography (OCT) image at the initial visit as intact (n = 26) and disruptive EZ (n = 12) groups, respectively. In addition, we classified the intact EZ eyes into clear (n = 15) and blurred (n = 11) EZ groups according to whether EZ on the SRF was observed distinctly or not. Multiple regression analyses showed that baseline EZ status significantly correlated (*p* = 0.0028) with the 12-months logarithm of the minimum angle of resolution (logMAR) best-corrected visual acuity (BCVA), indicating that baseline intact EZ significantly improves visual prognosis. The 12-months logMAR BCVA of the intact EZ group was significantly better (*p* < 0.001) than that of the disruptive EZ group, and did not differ significantly between the clear and blurred EZ groups. Thus, baseline foveal EZ status on vertical OCT images can be a novel biomarker for visual prognosis in eyes with SRF associated with BRVO.

## 1. Introduction

To date, studies have reported several factors affecting visual prognosis in branch retinal vein occlusion (BRVO) [1,2,3,4,5]. Younger patients [1,2,3,4] and a better baseline logarithm of the minimum angle of resolution (logMAR) best-corrected visual acuity (BCVA) [1,2,3,5] are good visual prognoses for BRVO.

Optical coherence tomography (OCT) findings have clarified factors affecting the visual prognosis for BRVO [3,5,6,7]. Intraretinal fluid and subretinal fluid (SRF) in eyes with BRVO are often detected using OCT images [8,9,10]. Although the ellipsoid zone (EZ) status is an important visual prognostic factor for BRVO [6,7], the effect of the presence of SRF on the visual prognosis for BRVO remains unclear [9,10,11,12]. We hypothesized that the presence of SRF does not affect the visual prognosis for BRVO, but that foveal EZ status on the SRF or other factors affect it in BRVO. However, few studies have been reported on visual prognostic factors limited to BRVO with SRF at the initial visit.

Therefore, we evaluated the initial OCT images of macular edema (ME) including SRF associated with BRVO, and examined whether foveal EZ on the SRF affects 12-months visual acuity.

## 2. Materials and Methods

### 2.1. Ethics

The study was conducted in accordance with the Declaration of Helsinki, and the protocol was approved by the Ethics Committee of the Shinseikai Toyama Hospital (number: 230127-3). We used an opt-out consent process and the requirement for informed consent was waived by the Ethics Committee of Shinseikai Toyama Hospital.

### 2.2. Patients

This retrospective observational study included patients with ME including SRF associated with treatment-naive BRVO. The medical and ocular histories of the patients were reviewed between 7 January 2017 and 29 December 2022 at the Department of Ophthalmology of the Shinseikai Toyama Hospital.

The inclusion criteria were as follows: untreated patients who had OCT B-scan images of ME including SRF at the initial visit within 6 months following BRVO onset and who received a single loading dose + pro re nata (one + PRN) regimen of 2.0 mg/0.05 mL aflibercept (Eylea^®^, Regeneron Pharmaceuticals, Inc., Tarrytown, NY, USA) over 12 months. Intravitreal injections of aflibercept were administered with one + PRN regimen when the central subfield thickness (CST) exceeded 300 μm.

The exclusion criteria were as follows:Presence of hemicentral retinal vein occlusion.Presence of significant media opacity that prevents evaluation of OCT findings.Presence of other retinal diseases, such as diabetic retinopathy or epiretinal membrane.Presence of severe glaucoma.Patients after vitrectomy (i.e., non-vitreous eyes).History of intra-ocular surgery (such as cataract and glaucoma surgery) 6 months prior to the study.Patients who underwent treatment, including intravitreal injection of steroids or other anti-vascular endothelial growth factor agents, or who were treated with a sub-Tenon injection of triamcinolone acetonide.Patients who could not perform adequately on the one + PRN regimen of aflibercept despite the presence of ME secondary to BRVO.Patients who were not followed for 12 months.

### 2.3. Examinations

All patients underwent a BCVA measurement using a Landolt C-chart, fundus camera (TRC-NW8, Topcon, Tokyo, Japan and/or California, Optos PLC, Dunfermline, UK), indirect ophthalmoscopy, slit-lamp biomicroscopy, and OCT (RS-3000, Nidek Co., Ltd., Gamagori, Japan and/or Spectralis OCT, Heidelberg Engineering, Heidelberg, Germany). Based on the vertical OCT images at the initial visit, SRF was defined as the vertical distance in the largest hypo-reflective space between the neurosensory retina and the inner line of the retinal pigment epithelium [13] and manually measured by one retinal specialist (H.S.) using the built-in software, and the SRF distance was automatically calculated in μm.

Perfusion status was evaluated by fluorescein angiography for all patients using an ultra-widefield fundus camera. A nonperfusion area smaller than five disc diameters was considered perfused [14]. Based on the previous study [15], the subtype of BRVO (i.e., major or macular) was also determined.

### 2.4. Definition and Classification of the Two Groups Using OCT

In the present study, we classified eyes with or without a continuous EZ band on the SRF of the central foveola on vertical OCT images through the fovea at the initial visit as the intact (Figure 1) and disruptive EZ (Figure 2) groups, respectively. At times, in the acute phase of BRVO, it is difficult to evaluate the EZ continuity from the affected side to the central foveola due to severe retinal hemorrhage and ME. Therefore, in this study, we assessed the foveal EZ band continuity on the SRF from the healthy side using vertical OCT images. The EZ band continuity was independently evaluated by two observers (H.S. and H.I.). In cases of any discrepancy, a third observer (M.Z.) made the final decision.

### 2.5. How to Divide Intact Ellipsoid Zone Band into Two Groups Using OCT

In eyes classified as intact EZ, there are two subcategories: those in which the EZ band is clearly observed on the SRF at the central foveola (clear EZ, Figure 3) and those in which there is no disruption of the EZ band but the EZ band is not clearly observed on the SRF at the central foveola (blurred EZ, Figure 4) as in central serous chorioretinopathy with elongation of the outer segment. In this study, in eyes classified into intact EZ group, the eyes with intact EZ bands were further classified into the clear and blurred EZ groups, respectively. Two observers (H.I. and H.S.) independently classified the eyes included in the two groups. In cases with a discrepancy, a third observer (M.Z.) made the final decision.

### 2.6. Assessment of the Foveal Ellipsoid Zone Band at Month 12

Using the vertical OCT image through the fovea at month 12, an observer (H.S.) evaluated the continuity of the foveal EZ band. If the continuity of the EZ band was ambiguous, another observer (M.Z.) made the final decision.

### 2.7. Statistical Analysis

Statistical analyses were performed using SAS version 9.4 software (SAS Institute, Inc., Cary, NC, USA). Data were expressed as the mean (standard deviation). The Shapiro–Wilk test was used to evaluate normality. For paired analyses, the Wilcoxon signed-rank test was used. For unpaired analyses, the Unpaired *t*-test was used. The changes in logMAR BCVA were determined as the 12-months minus baseline logMAR BCVA values. Fisher’s exact test was used in the categorical data between the two groups. Multiple regression analysis was performed to identify explanatory variables with a statistically significant contribution to 12-months logMAR BCVA. LogMAR BCVA at 12 months was set as the objective variable. Explanatory variables included patient age, groups (intact or blurred EZ), baseline logMAR BCVA, and baseline CST upon confirming the absence of multicollinearity among these explanatory variables. A *p*-value less than 0.05 was considered statistically significant.

## 3. Results

### 3.1. Patient Characteristics

Thirty-eight eyes from 38 patients met the study criteria for the analysis. Table 1 summarizes the patient characteristics. The logMAR BCVA and CST at month 12 showed significant improvement (*p* < 0.001 and *p* < 0.001, respectively) compared to the baseline values.

### 3.2. Comparison of the Two Groups According to Foveal Ellipsoid Zone Band Status at the Initial Visit

Table 2 summarizes the patient characteristics between the two groups. The concordance rate of classification into the intact and disruptive EZ groups by the two observers was 89.5% (34/38). At baseline and 12 months, the logMAR BCVAs were significantly better (*p* < 0.001 and *p* < 0.001, respectively) in the intact EZ group compared to that in the disruptive EZ group. The baseline CST in the disruptive EZ group tended to be higher than in the intact EZ group but did not differ significantly. The baseline SRF thickness was significantly higher (*p* = 0.019) in the disruptive EZ group than in the intact EZ group. The foveal EZ band was significantly more disrupted (*p* < 0.001) in the disruptive EZ (58.3%) group than in the intact EZ (3.8%) group at 12 months. One eye in the intact EZ group showed increased retinal hemorrhage during the follow-up periods and did not maintain the foveal EZ continuity at 12 months following the initial treatment (Figure 5). In the disruptive group, EZ continuity in five cases with EZ recovery at 12 months and the seven cases without EZ recovery are shown in Table 3.

### 3.3. Comparison of the Two Groups According to Whether the Intact Ellipsoid Zone Band Is Clear or Blurred at the Initial Visit

Table 4 shows a summary of the patient characteristics of the two subgroups of the intact EZ group (n = 26). The concordance rate for the classification of the intact EZ group into the clear and blurred EZ groups by the two observers was 88.5% (23/26). The baseline CST and baseline SRF thicknesses were significantly higher (*p* = 0.031 and *p* < 0.001, respectively) in the blurred EZ group than in the clear EZ group. The subretinal hemorrhage (SRH) was significantly more detected (*p* < 0.001) in the blurred EZ (90.9%) group than in the clear EZ (0%) group. However, the logMAR BCVAs at baseline and 12 months and the CSTs at 12 months did not differ significantly between the two groups.

### 3.4. Baseline Characteristics Which Were Associated with Visual Acuity at Month 12

Table 5 shows the results of the multiple regression analyses. The multivariate analyses demonstrated that the intact EZ group significantly correlated with the 12-months logMAR BCVA in eyes with ME including SRF secondary to BRVO (*p* = 0.0028, standardized coefficient = −0.51). However, in this study, patient age, baseline logMAR BCVA, and baseline CST did not correlate with the 12-months logMAR BCVA.

## 4. Discussion

In the present study, we examined the visual prognostic factors in eyes with ME including SRF secondary to treatment-naive BRVO using vertical OCT images. The 12-months logMAR BCVA was significantly better in the intact EZ group compared to that in the disruptive EZ group. Multiple regression analyses showed a significant correlation between baseline EZ status and 12-months logMAR BCVA. Meanwhile, a comparison of eyes in the clear and blurred EZ groups revealed no significant differences in the 12-months logMAR BCVA. These findings suggest that baseline foveal EZ status could be a novel biomarker for the visual prognosis in eyes with SRF secondary to BRVO.

In this study, the total number of injections needed for each patient was 4.2 (1.7), which was similar to those of previous studies [16,17]. The 12-months logMAR BCVA and CST improved significantly compared to the baseline levels. Although no significant differences in the total number of injections and CST at the 12 months between the intact and disruptive groups were observed, the 12-months logMAR BCVA in the intact EZ group was significantly better than that in the disruptive EZ group. These results suggest that baseline EZ status affects visual prognosis in eyes with SRF secondary to BRVO.

In eyes with SRF associated with BRVO, when the foveal EZ band on SRF is continuous, there are two patterns where the EZ band could be clearly observed or not. Therefore, in this study, the eyes with intact EZ were further classified into clear and blurred EZ groups based on the vertical OCT images. The results showed that the blurred EZ group had significantly higher baseline CST and baseline SRF thicknesses than the clear EZ group. Moreover, the SRH was significantly more detected in the blurred EZ group than in the clear EZ group. However, there were no significant differences in the 12-months logMAR BCVA between the two groups. These results suggest that the difference in the appearance of the clear and blurred EZ band on OCT images may be influenced by the CST and SRF thicknesses, and whether the baseline EZ band is continuous or not (i.e., intact or disruptive) influences the visual prognosis in eyes with SRF secondary to BRVO.

In addition, in eyes with SRF secondary to BRVO, we demonstrated that baseline EZ status was significantly associated with 12-months logMAR BCVA. This indicates that the visual prognosis is better in eyes with intact EZ than in those with disruptive EZ. In this study, foveal EZ band continuity at the 12 months following the initial treatment remained 96.2% and 41.7% continuous in the intact and disruptive EZ groups, respectively, which was significantly different between the two groups. The integrity of the photoreceptor layer in the fovea is presumably associated with visual acuity in resolved ME due to BRVO [18,19]. Therefore, the results of this study suggest that the baseline foveal EZ status on SRF affects the 12-months EZ status and 12-months logMAR BCVA. On the contrary, the baseline CST in the disruptive EZ group tended to be higher than that in the intact EZ group, and the baseline SRF thickness was significantly higher in the disruptive EZ group than in the intact EZ group. A thicker CST and SRF, which shows the severity of ME, could result in the disorganization of the baseline photoreceptor structure and photoreceptor dysfunction [20].

One eye in the intact EZ group showed increased retinal hemorrhage during the follow-up periods and did not maintain the foveal EZ continuity at 12 months following the initial treatment (Figure 5). Lee and Kim [21] reported that eyes with a ≥50% increased retinal hemorrhage are associated with poor visual prognosis of BRVO. Based on the results of this study, we believe that eyes classified into the intact EZ have a good visual prognosis, but close follow-up is needed because re-occlusion may occur.

Moreover, EZ continuity in five out of twelve eyes classified into the disruptive EZ group was recovered at 12 months. Iijima [22] reported that there are cases of EZ band recovery during the course of the disease. The detailed mechanism of EZ recovery remains unclear, but maintaining a low fluctuation in foveal retinal thickness may contribute to EZ band recovery [23]. This study did not identify any factors contributing to EZ band recovery (Table 3). This may be due to the small sample size. Further studies with a large number of cases are needed to identify factors that contribute to EZ recovery in the future.

Previous studies have described the patient’s age [1,2,3,4] and baseline BCVA [1,2,3,5] as potential visual prognostic factors for BRVO; however, the effect of CST on the visual prognosis in BRVO remains unclear [6,24,25,26]. In this study, the patient’s age, baseline logMAR BCVA, and baseline CST were not associated with the logMAR BCVA at 12 months. Since these results could contribute to the study design (i.e., this study included only eyes with SRF secondary to BRVO), sample size, and treatment strategies, prospective studies with uniform treatment strategies including a larger number of cases are needed to clarify the effects of these factors on the visual prognosis in eyes with SRF secondary to BRVO.

This study had several limitations. First, there was an inherent sampling bias due to the retrospective nature of the study. Second, during the study period, 20 eyes underwent scatter laser photocoagulation for the nonperfusion areas to prevent neovascular complications, such as vitreous hemorrhage, which might influence the results of this study. Although further studies are needed to confirm these results, our study focused on patients with ME including SRF associated with BRVO and showed that baseline EZ status can contribute to visual acuity at 12 months.

## 5. Conclusions

Baseline foveal EZ status on vertical OCT images can aid in the prediction of a visual prognosis in patients with SRF secondary to BRVO.

## Figures and Tables

**Figure 1 jcm-12-02909-f001:**
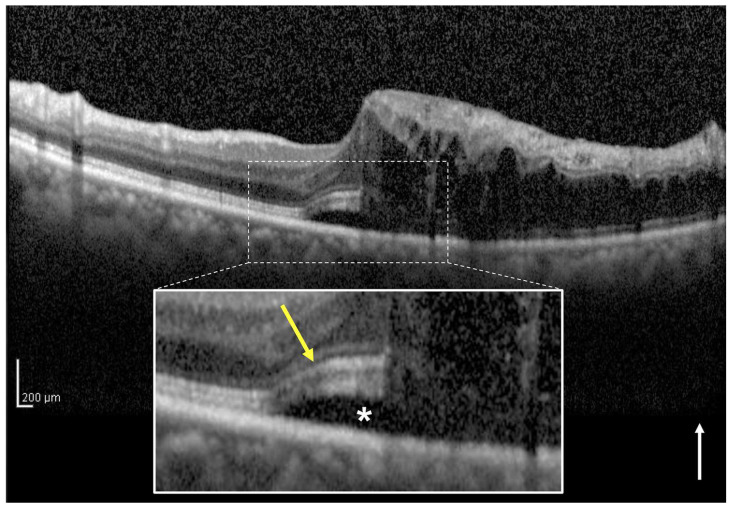
A vertical optical coherence tomography (OCT) image of the right eye with macular edema including subretinal fluid (SRF) associated with branch retinal vein occlusion in a 69-year-old woman. The vertical OCT image at the initial visit shows intraretinal fluid and SRF (asterisk). The white box below shows a magnified image of the foveola (dotted square). The ellipsoid zone (EZ) band (yellow arrow) from the healthy side appears continuous to the central foveola on the SRF. This eye was classified into the intact EZ group. The baseline best-corrected visual acuity of the patient was 0.4. The white arrow indicates the OCT scan direction.

**Figure 2 jcm-12-02909-f002:**
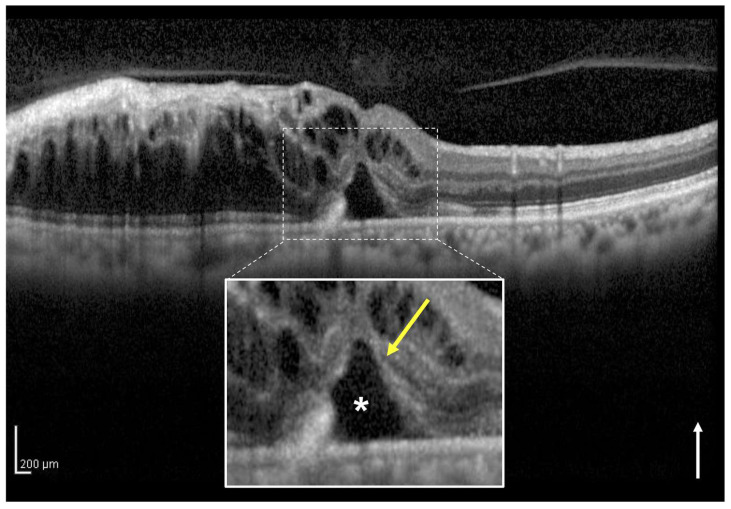
A vertical optical coherence tomography (OCT) image of the right eye with macular edema including subretinal fluid (SRF) associated with branch retinal vein occlusion in a 64-year-old woman. The vertical OCT image at the initial visit shows intraretinal fluid and SRF (asterisk). The white box below shows a magnified image of the foveola (dotted square). The ellipsoid zone (EZ) band (yellow arrow) from the healthy side does not appear continuous to the central foveola on the SRF. This eye was classified into the disruptive EZ group. The baseline best-corrected visual acuity of the patient was 0.2. The white arrow indicates the OCT scan direction.

**Figure 3 jcm-12-02909-f003:**
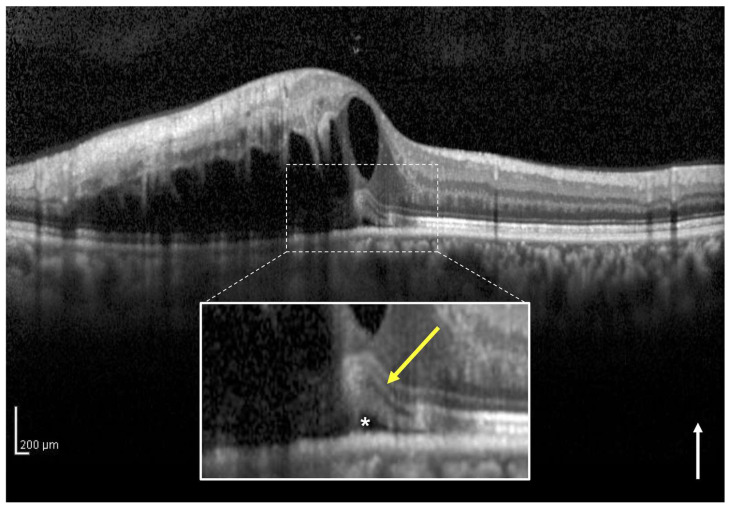
A vertical optical coherence tomography (OCT) image of the left eye with macular edema including subretinal fluid (SRF) associated with branch retinal vein occlusion of a 50-year-old man. The vertical OCT image at the initial visit shows intraretinal fluid and SRF (asterisk). The white box below shows a magnified image of the foveola (dotted square). The ellipsoid zone (EZ) band (yellow arrow) from the healthy side appears continuous and clear to the central foveola on the SRF. This eye was classified into the intact and clear EZ group. The baseline best-corrected visual acuity of the patient was 0.6. The white arrow indicates the OCT scan direction.

**Figure 4 jcm-12-02909-f004:**
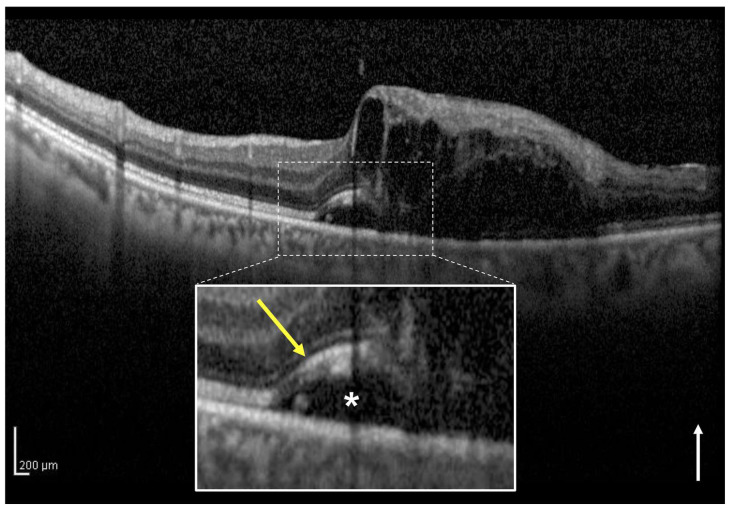
A vertical optical coherence tomography (OCT) image of the left eye with macular edema including subretinal fluid (SRF) associated with branch retinal vein occlusion of a 63-year-old man. The vertical OCT image at the initial visit shows intraretinal fluid and SRF (asterisk). The white box below shows a magnified image of the foveola (dotted square). The ellipsoid zone (EZ) band (yellow arrow) from the healthy side appears continuous but is blurred due to the outer segment elongation of the central foveola on the SRF. This eye was classified into the intact and blurred EZ group. The baseline best-corrected visual acuity of the patient was 0.5. The white arrow indicates the OCT scan direction.

**Figure 5 jcm-12-02909-f005:**
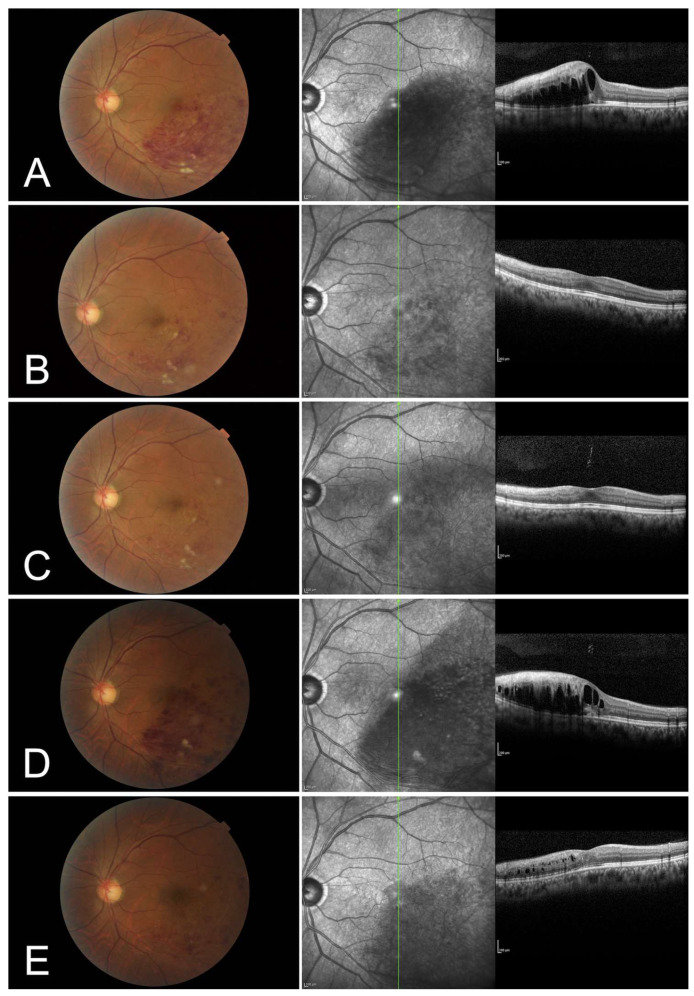
A 50-year-old man with branch retinal vein occlusion (BRVO) of the left eye with an ellipsoid zone (EZ) band on the subretinal fluid (SRF) was clearly observed at the initial visit but not at the 12 months following the initial treatment. (**A**) At the initial visit, the color fundus photograph shows retinal hemorrhage and cotton wool spots due to BRVO. The vertical optical coherence tomography (OCT) image demonstrates macular edema (ME) including SRF. The EZ band from the healthy side appears continuous to the central foveola on the SRF. The best-corrected visual acuity (BCVA) of the patient was 0.6. (**B**,**C**) At the 2nd (**B**) and 3rd (**C**) months following the initial treatment, the color fundus photographs show that retinal hemorrhage tends to decrease, and the vertical OCT images reveal that ME resolves following the intravitreal injections of aflibercept (**B**,**C**). (**D**) At 6 months following the initial treatment, the color fundus photograph shows increased retinal hemorrhage and the vertical OCT images reveal recurrent ME including SRF and subretinal hemorrhage. (**E**) At the 12 months following the initial treatment, the color fundus photograph shows that retinal hemorrhage tends to decrease. Although the vertical OCT images reveal that ME almost resolves following the additional intravitreal injections of aflibercept, the foveal EZ band appears disruptive. The 12-months BCVA of the patient was 0.5.

**Table 1 jcm-12-02909-t001:** A summary of the patient characteristics included in the present study.

Parameters	Value	*p*-Value
Total no. of eyes	38	
Baseline		
Age (years)	66.4 (12.7)	
Gender (male/female)	12/26	
Affected eye (right/left)	22/16	
Lens status (phakic/pseudophakic)	29/9	
Hypertension (%)	21 (55.3)	
Diabetes mellitus (%)	9 (23.7)	
Dyslipidemia (%)	8 (21.1)	
Subtype (major/macular)	30/8	
Perfusion status (ischemic/perfused)	25/13	
Duration before initial treatment (weeks)	5 (5.2)	
LogMAR BCVA	0.38 (0.32)	
Central subfield thickness (μm)	583.5 (133.3)	
Subretinal fluid thickness (μm)	155.9 (121.5)	
Intact EZ band/disruptive EZ band	26/12	
Subretinal hemorrhage (%)	18 (47.4)	
12 months following the initial treatment		
Total no. of injections	4.2 (1.7)	
Scatter laser photocoagulations during the follow-up periods (%)	20 (52.6)	
LogMAR BCVA	0.089 (0.29)	<0.001 *
Central subfield thickness (μm)	273.8 (29.2)	<0.001 *
Foveal EZ band disruption (%)	8 (21.1)	

* The Wilcoxon signed-rank test was analysed for comparisons between the parameters at the baseline and 12 months. BCVA, best-corrected visual acuity; EZ, ellipsoid zone; LogMAR, logarithm of the minimum angle of resolution; No., number.

**Table 2 jcm-12-02909-t002:** A summary of the patient characteristics of the two groups.

Parameters	Intact EZ	Disruptive EZ	*p*-Value
Total no. of eyes	26	12	
Baseline			
Age (years)	65.3 (12.4)	68.9 (13.6)	0.42
Gender (female)	18	8	1
Affected eye (right)	14	8	0.5
Lens status (pseudophakic)	7	2	0.69
Hypertension (%)	15 (57.7)	6 (50)	0.73
Diabetes mellitus (%)	8 (30.8)	1 (8.3)	0.22
Dyslipidemia (%)	6 (23.1)	2 (16.7)	1
Subtype (major)	19	11	0.39
Perfusion status (ischemic/perfused)	15/11	10/2	0.16
Duration before initial treatment (weeks)	4.5 (5.7)	6.1 (3.9)	0.39
LogMAR BCVA	0.23 (0.2)	0.7 (0.3)	<0.001
Central subfield thickness (μm)	556.2 (124.3)	642.8 (138.1)	0.062
Subretinal fluid thickness (μm)	125 (116.6)	223 (107.8)	0.019
Subretinal hemorrhage (%)	10 (38.5)	8 (66.7)	0.16
12 months following the initial treatment			
Total no. of injections	4.0 (1.7)	4.4 (1.8)	0.54
Scatter laser photocoagulations during the follow-up periods (%)	12 (46.2)	8 (66.7)	0.31
LogMAR BCVA	−0.045 (0.1)	0.38 (0.3)	<0.001
Change in logMAR BCVA	−0.27 (0.2)	−0.32 (0.4)	0.64
Central subfield thickness (μm)	274.9 (20.5)	271.4 (43.6)	0.74
Foveal EZ band disruption (%)	1 (3.8)	7 (58.3)	<0.001

BCVA, best-corrected visual acuity; EZ, ellipsoid zone; LogMAR, logarithm of the minimum angle of resolution; No., number.

**Table 3 jcm-12-02909-t003:** Comparison of the presence of EZ recovery at 12 months in the disruptive group.

Parameters	Recovered EZ	Non-Recovered EZ	*p*-Value
Total no. of eyes	5	7	
Age (years)	66 (12.0)	71 (15.1)	0.55
Duration before initial treatment (weeks)	6.2 (4.4)	6 (3.8)	0.93
Baseline logMAR BCVA	0.58 (0.3)	0.78 (0.4)	0.33
Baseline central subfield thickness (μm)	625.4 (44.0)	655.1 (182.4)	0.73
Baseline subretinal fluid thickness (μm)	201.4 (53.5)	238.4 (136.9)	0.58
Baseline subretinal hemorrhage (%)	3 (60)	5 (71.4)	1
Total no. of injections	4 (1.4)	4.7 (2.1)	0.53

BCVA, best-corrected visual acuity; EZ, ellipsoid zone; LogMAR, logarithm of the minimum angle of resolution; No., number.

**Table 4 jcm-12-02909-t004:** A summary of the patient characteristics of the two subgroups of the intact ellipsoid zone.

Parameters	Clear EZ	Blurred EZ	*p*-Value
Total no. of eyes	15	11	
Baseline			
Age (years)	66.3 (11.8)	63.9 (13.7)	0.63
Gender (female)	10	8	1
Affected eye (right)	9	5	0.69
Lens status (pseudophakic)	3	4	0.41
Hypertension (%)	9 (60)	6 (54.5)	1
Diabetes mellitus (%)	5 (33.3)	3 (27.3)	1
Dyslipidemia (%)	4 (26.7)	2 (18.2)	1
Subtype (major)	12	7	0.41
Perfusion status (ischemic/perfused)	11/4	4/7	0.11
Duration before initial treatment (weeks)	3.3 (4.5)	6.1 (6.9)	0.23
LogMAR BCVA	0.23 (0.2)	0.22 (0.2)	0.95
Central subfield thickness (μm)	511.9 (100.1)	616.5 (132.8)	0.031
Subretinal fluid thickness (μm)	45.5 (28.7)	233.4 (102.1)	<0.001
Subretinal hemorrhage (%)	0 (0)	10 (90.9)	<0.001
12 months following the initial treatment			
Total no. of injections	4.1 (1.7)	4.0 (1.7)	0.92
Scatter laser photocoagulations during the follow-up periods (%)	7 (46.7)	5 (45.5)	1
LogMAR BCVA	−0.03 (0.1)	−0.066 (0.1)	0.39
Change in logMAR BCVA	−0.26 (0.2)	−0.29 (0.2)	0.65
Central subfield thickness (μm)	274.4 (21.7)	275.6 (19.7)	0.88
Foveal EZ band disruption (%)	1 (6.7)	0 (0)	1

BCVA, best-corrected visual acuity; EZ, ellipsoid zone; LogMAR, logarithm of the minimum angle of resolution; No., number.

**Table 5 jcm-12-02909-t005:** Baseline visual prognostic factors affecting the 12-months visual acuity in the multiple regression analysis.

Explanatory Variables	Coefficient (95% CI)	*p*-Value	Standardized Coefficient
Intact EZ (compared to disruptive EZ)	−0.31 (−0.5 to −0.11)	0.0028	−0.51
Age (years)	0.004 (−0.002 to 0.001)	0.15	0.18
Baseline logMAR BCVA	0.14 (−0.16 to 0.44)	0.35	0.16
Baseline central subfield thickness (μm)	0.0004 (−0.00007 to 0.0009)	0.09	0.2

BCVA, best-corrected visual acuity; CI, confidence interval; EZ, ellipsoid zone; LogMAR, logarithm of the minimum angle of resolution.

## Data Availability

The datasets used in this study are available from the corresponding author upon reasonable request.

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
