# Peer review of "Visual Prognostic Factors in Eyes with Subretinal Fluid Associated with Branch Retinal Vein Occlusion"

_jcm, 2023, doi:10.3390/jcm12082909_

Round 1
Reviewer 1 Report
This study aims to evaluate vertical OCT images of macular edema including subretinal fluid (SRF) secondary to BRVO and investigated whether foveal ellipsoid zone (EZ) on the SRF using vertical OCT images at the initial visits affects visual prognosis.
I think that SRF secondary to BRVO in the initial OCT has less impact on the EZ damage. If the SRF remains long term, I believe it will change the EZ statas. We do not find any novelty in this study because it is obvious that EZ damage with or without SRF on initial OCT is related to visual prognosis.
Reviewer 2 Report
It was a great pleasure for me to review you paper "Visual Prognostic Factors in Eyes with Subretinal Fluid Associated with Branch Retinal Vein Occlusion". It is extremelly clear and well written. I would suggest some increments that may benefit the paper.
- Would you consider to classify the EZ disruption in the entire lines (high definition) and not only in the central one?
- All the tables: line of "Baselin" and "12 months following treatment" should be placed in the right side, in order to diferentiate from the others.
- Table 2: Perfusion status (ischemia) - better to see Perfusion status (ischemia/ perfused); LogMAR at 12 months - please add a line with change in logmar BCVA, since the initial BCVA was not the same - the same for table 3.
- Discussion: After line 271 - Please also explain why there was resolution of disruption in 5/12 eyes after 12 months. Did you found any factor responsible for it? Was it predictive of a better VA outcome?
Thank you so much for you input.
Reviewer 3 Report
The authors investigated whether the status of EZ on the subretinal fluid affected visual prognosis at 12 months after the initial treatment in the 38 eyes with BRVO received intravitreal aflibercept injections. They found that the good integrity of EZ was a significant positive factor to achieve better visual prognosis. They suggest that baseline foveal EZ status on vertical OCT images could be a novel biomarker for the visual prognosis in eyes with SRF secondary to BRVO.
Major
I can’t find any description on figure 5 in results.
The authors performed IVA treatment with PRN regime, however, the criteria of injection was not shown. It is unclear whether the same criteria were used for treatment in the two groups.
Minor
Need English edition. The followings are examples.
Line 211, Baseline characteristics which were associated with visual acuity at months 12.
Line 266, The meaning of this sentence is unclear.
Line 267, The eyes
Round 2
Reviewer 1 Report
Corrected correctly and no problem.
Reviewer 3 Report
I confirmed the authors' revision appropriate.